# Sustainable Development in the Agri-Food Sector in Terms of the Carbon Footprint: A Review

**Magdalena Karwacka** **, Agnieszka Ciurzyńska, Andrzej Lenart** **and Monika Janowicz \***

Faculty of Food Sciences, Department of Food Engineering and Process Management, Warsaw University of Life Sciences, SGGW, 02-787 Warszawa, Poland; mkarwacka@vp.pl (M.K.); agnieszka_ciurzynska@sggw.edu.pl (A.C.); andrzej_lenart@sggw.edu.pl (A.L.)
*   Correspondence: monika_janowicz@sggw.edu.pl; Tel.: +48-022-59-37566

**Abstract:** The concept of sustainable development is increasingly important in the agri-food sector and global economy. International activities are undertaken to improve the efficiency of industry by reducing its negative impact on the environment. To help determine harmful human activity, the environmental footprints of products and services are calculated using the LCA (life cycle assessment) method. The purpose of this article was to explain topics of sustainable development and environmental footprints, especially the carbon footprint in the agri-food sector, based on the latest literature. The agri-food industry consumes around 30% of global energy demand. It is also a source of emissions of a significant part of greenhouse gases released into the environment. The carbon footprint of food products is determined by many factors associated with their production. Food of animal origin is more harmful and has higher carbon footprints than plant-based products. GHG emission reduction is possible due to the use of renewable energy sources and the abandonment of the use of artificial fertilizers and plant protection products.

**Keywords:** carbon footprint; food; sustainable development; agri-food sector; environmental footprints

## 1. Introduction

The agri-food sector includes two inseparable sectors of the economy: agriculture, which is a source of plant and animal raw materials, and food processing, which is the main recipient of agricultural crops, also responsible for stimulating and directing the production of agricultural raw materials. The term food industry refers to the production of food, beverages, and tobacco. It applies to all activities related to the production, processing, distribution, preparation, and consumption of food, taking into account socio-economic and environmental aspects. In Poland, the agri-food industry has been in a continuous development phase for several decades. Poland's accession to the European Union was of great importance in this matter, which resulted in deep restructuring and modernization of agriculture, related to subsidies enabling adaptation of production enterprises to the standards in force in the EU. Thanks to this, the Polish agri-food sector has become competitive both on the domestic and international market [1–4].

Currently, the development of the agri-food industry strives for complete automation of production processes. Intelligent greenhouses, robots, drones, intelligent sensor networks, and closed production systems in which people do not directly participate are increasingly used. It is characterized by the implementation of various tools enabling the digitization of food production systems, the pursuit of maximum reduction in labor costs while maintaining the quality and safety of manufactured products, as well as the introduction of sustainable development principles by reducing the consumption of water, fuels, and fertilizers, and promoting the use of renewable energy [5–7].

The main goal of the aforementioned concept of sustainable development is to search for effective ways to solve the problems of the population around the world, including meeting their current needs, without limiting the possibility of meeting these needs for future generations. It addresses the challenges of increasing energy demand, climate change, environmental pollution, migration of people, ensuring food safety, and many other issues, the solution of which requires the cooperation of representatives of the world of science, politics, and economy. The development of sustainable development indicators, based on comparative analyzes of actual data on products, enterprises, and investments, and their consistent application is necessary to reduce the degradative impact of human activities on the environment [8,9].

In Poland, sustainable development policy is a constitutional principle. Article 5 of the Constitution of the Republic of Poland of 2 April 1997 states that "the Republic of Poland shall safeguard the independence and integrity of its territory and ensure the freedoms and rights of persons and citizens, the security of the citizens, safeguard the national heritage, and shall ensure the protection of the natural environment pursuant to the principles of sustainable development". A reference to the idea of sustainable development is also found in article 74, relating to ecological security and the principles of generational justice and environmental protection. The quoted provisions in the Constitution of the Republic of Poland impose on public authorities the obligation to conduct a policy ensuring ecological security for current and future generations. The most important legal acts, taking into account the concept of sustainable development, that are or were in force in Poland include: The Energy Law of 10 April 1997, as amended, The Environmental Protection Law of 27 April 2001, Resolution No. 163 of the Council of Ministers of 25 April 2012 on the adoption of the "Strategy for the sustainable development of rural areas, agriculture, and fisheries" for 2012–2020, and Resolution No. 123 of 15 October 2019 on the adoption of the "Strategy for the sustainable development of rural areas, agriculture, and fisheries" [10,11].

In 2015, 193 Member States of the United Nations, including Poland, adopted the document entitled "Transforming our world: the 2030 Agenda for Sustainable Development", on which work began three years earlier at the United Nations Conference on Sustainable Development. This document defines 17 goals and 169 tasks, covering a wide range of environmental, social, and economic issues, including climate change, energy demand, biodiversity, food supply and security, sustainable production, and consumption, as well as health care, education, gender, equality, peace, and economic growth [12]. Based on the above-mentioned agenda, the Polish government has developed a "Strategy for the sustainable development of rural areas, agriculture, and fisheries", which was adopted by the Council of Ministers together with Resolution No. 123 of 15 October 2019 [13].

The most important activities related to the agri-food sector include: reducing overproduction and food waste, ensuring equal access to drinking water and food for all people, promoting sustainable consumption and economic development, reducing emissions of pollutants to the atmosphere, water, and land, and sustainable management of natural resources. Food is a source of threats to the principles of the concept of sustainable development; it is related to the impact on social relations and their economic conditions. Above all, however, the key role is played by the issues related to the impact on consumer health, inter alia, through the impact on the surrounding environment. Agricultural production takes place on almost 40% of the world's land area. With such an extensive area, 70% of fresh water is used. It should also not be forgotten that the agri-food industry is the basic industry in most countries of the world and is the center of its global economy with an estimated value of billions of dollars per year. The research shows that this branch of the world economy is developing dynamically, operating within the socio-economic system. For its proper functioning, it uses natural resources (e.g., water and land demand), the economic system (e.g., a network of suppliers, processors, and subcontractors), and social systems, which include consumer organizations. The entire agri-food industry is based on a network of connections between agriculture, food processing, and trade, which are internal subjects of this system. On the other hand, the external basis of this system are consumers, governmental and non-governmental organizations, as well as research and financial

institutions, which also affect the functioning of the agri-food sector. The way food is produced, processed, distributed, and consumed is influenced by global and local trends such as urbanization, industrialization, cultural and demographic changes, and climate change. Therefore, cooperation and effective communication of all links in the food chain is the key to improving the operation of the entire system and effective implementation of the sustainable development policy [14,15].

In connection with the above, several directions of the evolution of the agri-food industry can be proposed, including the principles of sustainable development in this sector in a more determined manner. One of the ideas of modification in the agri-food industry is production optimization, i.e., the implementation of techniques and technologies that support and intensify the course of processes in the social, economic, and environmental structure. The search for the possibility of using renewable energy sources (RES) is an important aspect of the idea of sustainable development included in the policy for both the European Union and individual member states. This is due to the fact that the use of conventional energy sources causes environmental damage related to both the exploitation and depletion of mineral resources, and their use results in the emission of pollutants to the environment. The growing popularity of RES results not only from their ecological advantage, but also from economic conditions. On the one hand, this is the result of an increase in the prices of conventional energy carriers, as their exploitation from less and less accessible places contributes to a significant increase in extraction costs, and on the other hand, the improvement of technologies used for renewable energy sources, which also contributes to the reduction of waste generation. The creation of modern installations for the production of energy from wind, water, sun, and biomass sources have their benefits both at the local and global level, including in the agri-food sector. They can contribute to the economic growth of a given region, including by creating new jobs and significantly reducing greenhouse gas emissions, thus reducing the impact of energy on global climate change [16,17]. The implementation of international obligations resulting from the United Nations Framework Convention on Climate Change and the Kyoto Protocol to this Convention, regarding the reduction of $CO_2$, stimulates the development of green energy [18]. With the emergence of new technologies related to RES, especially with biomass, it is necessary to conduct a comprehensive environmental impact assessment and determine the environmental effects obtained related to the modernization of these systems [16,19]. Introducing quantification of environmental benefits would facilitate the monitoring of the environmental performance of an installation and a more effective comparison of different alternative scenarios. A method that meets the above criteria is LCA (life cycle assessment), which is a widely used technique of environmental management in the EU. It enables a holistic assessment of products, processes, and systems, i.e., taking into account the pre- and post-production phases; it also allows for quantifying the ecological effect in the form of one number—an eco-indicator. It contributes to a much easier interpretation of the results, as well as to a more efficient juxtaposition of two different technologies or systems [20]. Other promising results that make the agri-food sector part of the principles of sustainable development can be pursued by reducing the harmful impact on the environment by increasing the efficiency of using raw materials, energy, and other resources. As a result of the optimization and development of innovative products, their impact on people's quality of life in the field of health, education, and culture is changing at the employee, consumer, and community level. In economic terms, the implementation of the concept of sustainable development should result in: increased efficiency, the creation of cheap, high-quality products, and the creation of enterprises, and hence the creation of new jobs. The principles of sustainable development should be taken into account already at the stage of designing products and processes. Optimization includes the use of materials that have the lowest possible impact on the environment through the emission of pollutants and greenhouse gases, and their production does not require excessive water or energy resources. In addition, it is important to use the so-called clean technologies in production and packaging processes and enable reuse, recycling, or ecological disposal of waste (Figure 1).

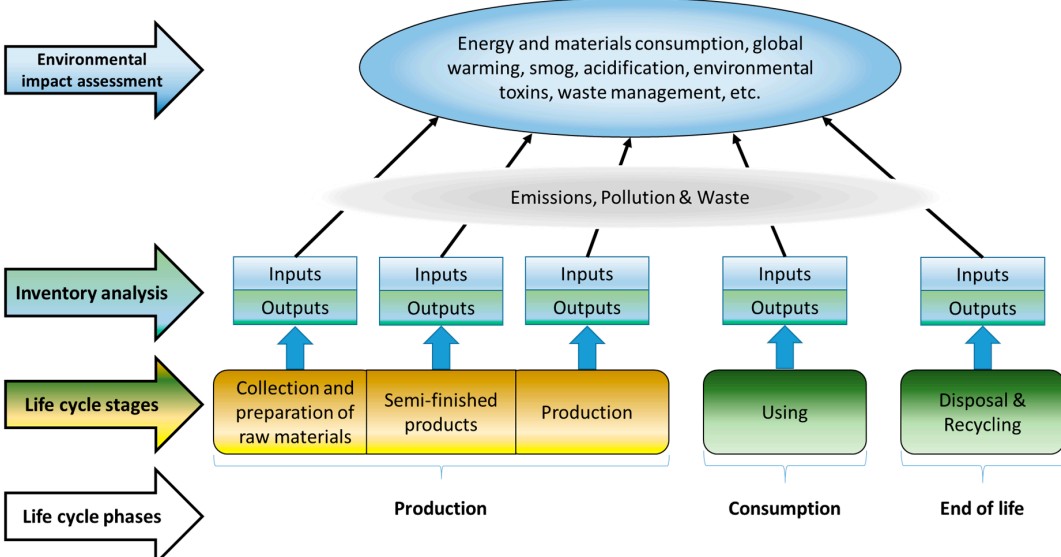

**Figure 1.** The life cycle range for the various assortments of the agri-food sector (own study based on [16]).

Taking into account the above and the importance of the environmental footprint in shaping the global economy, it should be presented to what extent the factors related to the agri-food sector and LCA determine the importance of the environmental footprint in the assessment of compliance with the principles of sustainable development (Figures 1 and 2).

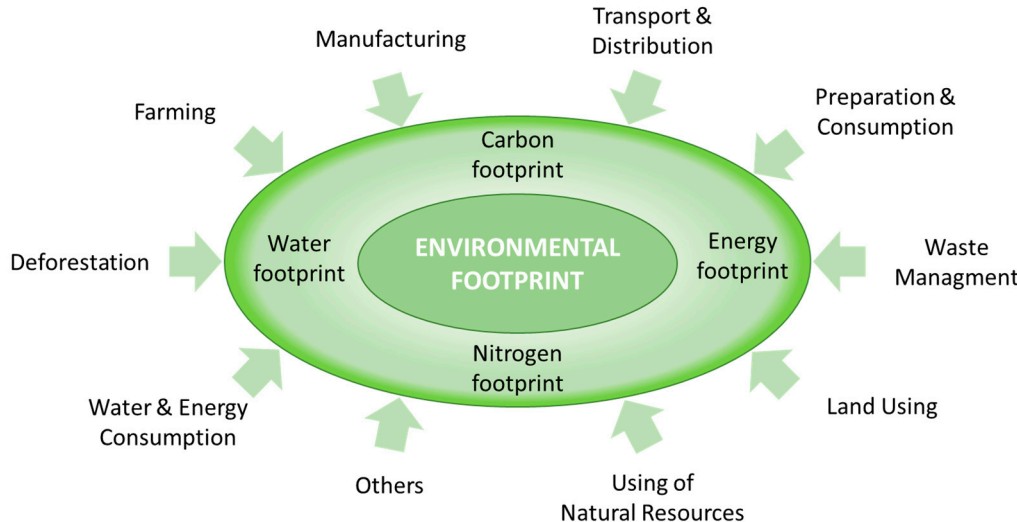

**Figure 2.** Determinants of the environmental footprint in the agri-food sector [own study].

## 2. Life Cycle Assessment (LCA) as a Method of Determining the Environmental Footprint

Research conducted in 2011 showed that food production is responsible for almost 30% of global energy consumption [21]. In addition to the significant amount of energy required for the proper functioning of all links in the production and distribution chain, food systems are responsible for one third of greenhouse gas (GHG) emissions such as carbon dioxide ($CO_2$), methane ($CH_4$), and nitrogen dioxide ($NO_2$) that directly affect climate change. The agri-food sector is one of the most emissive in the economy; it comes with about 57% of methane and as much as 90% of nitrogen dioxide [22]. According to the data collected by the National Center for Emissions Management in 2016, agriculture in Poland was the source of about 8.1% of total greenhouse gas emissions, including 77.6% nitrogen

oxide, 29.6% methane, and 0.32% carbon dioxide [23]. Therefore, much attention is paid to research on energy consumption and greenhouse gas emissions in food production. Developing a strategy to mitigate climate change involves, among other things, analyzing the life cycle stages of products that can be modified to reduce their environmental impact as much as possible (Figure 2) [24]. GHG is reduced by producing renewable energy as a substitute for fossil fuels, by reducing fugitive greenhouse gas emissions from fertilizers stored and landfilled, and by reducing the use of chemical fertilizers in plant production. Anaerobic digestion (AD) biogas produces biogas at an average rate of 0.30, 0.25, and 0.48 L/g of volatile solids from pig, cattle, and poultry slurry, respectively. The biogas produced is of high quality and contains 60–80% CH4. Wastewater from AD is better balanced to meet the needs of the crops than crude slurry, reducing the need for supplemental chemical nitrogen and phosphorus fertilizers. Both obtaining energy and reducing the need for chemical fertilizers will significantly reduce the carbon footprint of livestock foods.

On-farm biogas production contributes to a more sustainable livestock activity by significantly reducing other environmental impacts related to manure management. It reduces the risk of water contamination associated with animal slurry by removing 0.80–0.90 soluble chemical oxygen demand. This reduction allows for a more frequent and better timing of manure application. Both the duration of application and a better nutrient balance have the potential to increase plant nutrient uptake and minimize nutrient loss to the environment. Reducing the viability of weed seeds reduces the need for herbicides and makes bioreactor wastewater more acceptable to organic farmers [25,26].

Direct and some indirect environmental factors such as energy consumption, carbon dioxide emissions, the use of water and other resources including the extraction and processing of raw materials, their production, transport, use phases, as well as decommissioning are a practical method for assessing the life cycle of products of various types and origin (LCA—life cycle assessment). LCA methods are widely used to calculate "environmental footprints" of food products taking into account their entire life cycle [27]. The LCA study requires tracing the product path "from field to plate", that is, analysis of all production stages and operations that do not directly affect the production process, but are necessary to obtain the finished product [28]. Plant production includes the use of agricultural vehicles, the production of fertilizers and pesticides, emissions from soil, processing, transport, and utilization of waste [29]. When analyzing the production of animal products, the production of feed, milk, meat, eggs, manure management, slaughter, and waste utilization should be taken into account. Animal production consumes many more resources, such as land, water, and energy, and is a source of a significant part of biological pollution, which contributes to the degradation of the ecosystem [30]. The indicators, calculated as part of the life cycle assessment of products, are useful for comparing the environmental impact of foods with different nutritional values (Figure 1) [22].

Environmental footprints describe the environmental impact of a product or service. They are calculated for many goods and services, in particular for plant and animal products, but it is not common to include this type of information on food labels. The analysis of energy consumption and greenhouse gas emissions in the life cycle assessment of products provides estimated data, which is usually referred to as environmental footprints calculated per unit of food produced. These indicators make it possible to assess the efficiency of worldwide industry, including the agri-food sector (Figure 2) [22,31].

The most important and most frequently marked environmental footprints include: carbon footprint, water footprint, nitrogen footprint, and energy footprint. All these indicators are referred to as the environmental footprint. Other important issues that help assess the environmental impact of food products are also regarding pesticide use, the welfare of farmed animals, and the degree of deforestation. By combining the above data, it is possible to obtain information about the extent to which the manufacturing process of a particular food product affects the environment. Preliminary studies have shown that the mentioned footprints complement each other, and their combination can help determine the environmental footprint of humanity and contribute to reducing greenhouse gas emissions in accordance with the concept of sustainable development (Figure 2) [22,29,32].

### 2.1. Carbon Footprint (CF)

In the scientific literature, the concept of carbon footprint (CF) appeared in the 1960s in connection with the growing interest in climate change, which became increasingly noticeable. This term quickly spread to the political, business, and media arena around the world, because it was clearly associated with concern for the environment [33]. Despite the fact that the carbon footprint is a concept that has been widely used for several decades, there are still discussions about its correct definition. According to the Kyoto protocol, CF is considered to be the total amount of $CO_2$ equivalent and other greenhouse gases coming from the product's life cycle, including its storage, use, and disposal [34].

According to the Regulation of the European Parliament, a ton of $CO_2$ equivalent is the amount of greenhouse gases expressed as the product of the greenhouse gas mass in tonnes and their global warming coefficient [35]. This means that the unit of measure of the carbon footprint determines the emission of carbon dioxide, nitrous oxide, and methane, as well as hydrofluorocarbons, perfluorocarbons, sulfur hexafluoride, and other greenhouse gases [28,32]. Not all greenhouse gases affect climate change in the same way, which is why, in order to easily compare the carbon footprints of various products, they are converted to the amount of carbon dioxide using appropriate factors (e.g., methane impacts the environment 25 times more than $CO_2$, and nitrogen up to 298 times) [36].

### 2.2. Water Footprint (WF)

The water footprint indicator (WF) not only concerns the amount of water directly contained in the product, but also includes the water used during the processes of obtaining products and services. Therefore, the concept of water footprint is closely related to the term virtual water, i.e., water not physically present in the product, but used to make it. These concepts developed at the turn of the 20th and 21st centuries in connection with the shrinking water resources in the world [37]. For example, the amount of water used in meat production is 4.3 t·(kg of poultry$^{-1}$), 5.9 t·(kg of pork$^{-1}$), and 15.4 t·(kg of beef$^{-1}$), while the water footprint of basic plant materials (fruit, roots, and vegetables) does not exceed 1 t·(kg$^{-1}$) [38].

In general, the water footprint is a measure of the consumption and degradation of fresh water. The total water footprint indicator consists of three smaller ones. The first—green—relates to natural water, supplied in the form of precipitation and used by plants, the second—blue—refers to the use of groundwater and surface water, while the third—gray—measures the volume of water needed to assimilate produced pollutants. Initial studies included the consumption of green and blue water, but both types were treated as one, while gray water was not included at all. For the first time, three types of water footprints under the respective names were presented in 2008 by Hoekstra and Chapagain [39].

In research, there is another water footprint indicator, namely water use efficiency (WUE). It is defined as the ratio of yields to water used per crop. For wheat cultivation, both WUE and water footprint were determined, each over 35 years [40]. A non-linear relationship was found between these indicators due to a significant increase in the use of gray water. In both cases, the initial measurement assumptions are different. Thus, the water footprint covers not only direct but also indirect water consumption and its environmental impact. As a result, it shows a more comprehensive assessment of the water consumption of crops. Changes in the water footprint of crops of various crops, i.e., wheat, rice, cotton, and rape, have been studied over the course of years [41]. Rice cultivation is characterized by the greatest decrease in WF. In the case of wheat, a lower water consumption was also recorded, but the decrease is not so spectacular. The cultivation of cotton and rape only slightly decreased water consumption. Changes in these indicators result from the intensification of the urbanization process by disrupting the natural water cycle.

The interest in determining the water footprint of finished products is observed in many areas of the agri-food industry. Since 2008, the company Barilla [42,43] have begun to assess the environmental burden of its products using the life cycle assessment (LCA) methodology. Their aim was to cover the entire production chain and reduce the environmental impact of its products. The research was carried out in a systematic manner, thanks to which the company was able to calculate and verify the

main indicators of the environmental impact of its products. Their research focused in particular on the water footprint of Barilla's durum wheat pasta and the water associated with the durum wheat and pasta trade. It was found that the wheat cultivation stage had the largest share in the total water footprint of pasta. According to studies, the cultivation of cereals is characterized by a high value of the gray water footprint [42].

The growing demand for finished products is also associated with the increased use of water resources for their production. Consequently, the food industry has been forced to adopt an appropriate water management approach to reduce water demand. Water footprint data for ready meals, such as chilled vegetable soup (gazpacho), are presented in [44]. The determined total value of the WF parameter for 1 L of gazpacho was 580 L. For this process, the following water footprints were distinguished: green, blue, and gray, which were 69%, 23%, and 8%, respectively, and were related mainly to the supply chain, while the share of the operational water footprint was small [43,44].

## 2.3. Nitrogen Footprint (NF)

The nitrogen footprint (NF) is defined as the amount of reactive nitrogen released into the environment as a result of the unit's resource consumption and associated production [45].

In the last century, changes in the natural nitrogen cycle were observed as a result of human activities related to the "Green Revolution" and the need to use artificial fertilizers, most of which contain nitrogen. In the early 1960s, plants processed about 68% of the nitrogen compounds supplied to them as fertilizer, while in 2014 only 47%. This decrease shows that more than 50% of this element is lost to the environment, which contributes to progressive soil acidification, reduction of biodiversity, and increased greenhouse gas emissions [46]. Only 25% of nitrogenous compounds are of natural origin, while 75% are generated by human activities. However, it turns out that excessive concentration of nitrogen negatively affects not only the natural environment, but also the functioning of people. It is estimated that the economic losses associated with excessive nitrogen emissions in the European Union are in the range of EUR 70 to 320 billion per year. In addition, in 2012, approximately 72,000 premature deaths in the EU were caused by an increase in the amount of nitrogen dioxide in the atmosphere [47].

The use of nitrogen is inevitable, but in view of the challenges of food safety, environmental degradation, and climate change, the trade-off between the use of reactive nitrogen compounds and their remaining in the environment can be mitigated by increasing the efficiency of these compounds, reducing food waste, and the consumption of animal products [48].

Nitrogen (N) is an essential element for plants and animals. Due to the large inputs of mineral fertilizers, crop yields and livestock production in Europe have increased significantly over the last century, but as a consequence, losses of reactive air, soil, and water have intensified. Two different models (CAPRI and MITERRA) were used to quantify agricultural nitrogen flows in the European Union (EU-27), at country level and across EU-27 agriculture, broken down into 12 main food categories. The results showed that the N footprint, defined as total environmental loss per product unit, varies significantly between the different food categories, with much higher values for animal products and the highest for beef (approx. 500 g $N \cdot (kg\ beef)^{-1}$) compared to plant products. The lowest N trace of approx. 2 g $N \cdot (kg)^{-1}$ of product was calculated for sugar beet, fruit and vegetables, and potatoes. The reactive N losses were dominated by N leaching and run-off and ammonia volatilization, with 0.83 and 0.88 due to consumption of animal products. The investment factor N, defined as the amount of new reactive N needed to produce one unit of N in the product, ranged from 1.2 kg $N \cdot (kg\ N)^{-1}$ in the product for legumes to 15–20 kg N for beef [49].

## 2.4. Energy Footprint (EF)

In recent years, global energy production has undergone constant transformation. It is forecasted that energy consumption will increase every year, which will directly affect environmental degradation. To meet the growing demand, it will be necessary to intensify the exploitation of energy sources, which are inevitably coming to the end. That is one of the reasons why rich natural resources are

one of the major causes of armed conflict. At the same time, recognizing the potential social and biological consequences of climate change is a reason for pressure to regulate greenhouse gas emissions. Therefore, due to the desire to maintain energy security, as well as in accordance with the growing ecological awareness, the trend is taking place around the world for the use of renewable energy sources such as wind, hydro, and nuclear power plants [50].

The energy footprint (EF) defines the direct and indirect energy resources used to produce goods and services. It takes into account the production and consumption of energy through the entire supply chain, regardless of the place of production and distribution. The energy footprint can be determined at global, national, regional, local, industrial, and product levels. For this purpose, data on the life cycle of products and industrial and commercial chains are used [51]. The concept of "food miles" is directly related to the energy footprint. It determines the distance that the food must travel from the production site to the consumer's table. The amount of GHG emissions resulting from the transport of food largely depends on the means of communication used and is proportional to the distance traveled, i.e., the number of food miles. From an ecological point of view, air transport is the most burdensome for the environment, while land transport is the least. One of the most effective methods proposed to reduce the harmful effects of food production on the environment is to use local raw materials and simplify the distribution chain [52]. The ever-expanding structure of the global economy is of great importance in shaping the energy footprint. Cross-sectoral interactions and the increasing complexity of distribution systems contribute to the increase in energy consumption, and the growing network of economic connections makes it difficult to determine the impact of production on the environment [53].

### 2.5. Carbon Footprint of Plant and Animal Products

The topic of environmental footprints and sustainable development is becoming more and more popular in the world of science and politics, so it is important to study this issue as thoroughly as possible. In recent years, more and more research has been conducted to determine the traces of various types of products and services. According to FAO data, animal production, i.e., meat and dairy, accounts for 18% of global greenhouse gas emissions. It is estimated that a meatless diet is several times less harmful to the environment than a meat diet, depending on the type of meat consumed [54]. There are not many studies on consumer interest in products with a lower carbon footprint, but available sources show that this one exists. In general, people who attach importance to ecological aspects more appreciate the quality and added value of environmentally friendly production. Such consumers are also more likely to pay a higher price for products with a lower carbon footprint, in contrast to people with a more economical approach, for whom the product's environmental impact is not an argument to buy a more expensive but more ecologically produced item. However, this does not change the fact that, according to the respondents' preferences, the ideal situation would be to combine ecology with economy and offer environmentally friendly products at low prices [55].

In the past decade, the European Commission has conducted a number of activities aimed at developing uniform methods for measuring the environmental impact of products and enterprises from various industries. For the food sector, pet and farm animal feed, pasta, bottled water, dairy products, wine, beer, olive oil, coffee, sea fish, and meat were included. The main tool used in the research was life cycle assessment. Finally, methods of calculating the environmental footprint were adopted for several of the mentioned products for all types of feed, pasta, water, dairy products, wine, and beer. In the near future, manufacturers will be required to include environmental footprint information on labels to give consumers the opportunity to make an informed choice among products that affect the environment to varying degrees [56].

New Zealand scientists studied the formation of a carbon footprint of cow's milk from medium-sized farms in 2010–2018. The differentiating factors were time, region, and cattle breeding system. A slight downward trend was observed in the average values of the carbon footprint, which in the 2010/2011 season was 0.81 kg $CO_2 \cdot$(kg milk$^{-1}$), and after eight years it was equal to 0.78 kg $CO_2 \cdot$(kg milk$^{-1}$). Regional differences have also been recorded. It turned out that milk from the

region with more difficult weather and soil conditions (Northland) had an average greenhouse gas emission rate (GHG) that was higher by 6% on average. Over 70% of GHG emitted was methane, which is a product of cows' digestive processes. When developing the results, New Zealand-specific nitrous oxide emission factors were used, based on many validated field trials, resulting in 18% lower carbon footprint than when using the default indicators of the Intergovernmental Panel on Climate Change [57]. For comparison, the production of unsweetened almond milk in California has a carbon footprint of 0.71 kg $CO_2 \cdot$1.4 L container$^{-1}$. In this case, the largest emission (46% of total) was recorded during packaging and transport [58].

In order to examine the influence of the raw material characteristics on the product's carbon footprint, the indicators of cow's cheese produced in a traditional Spanish factory were compared. The cheese was made from milk from cows from pasture-based and semi-confinement systems. In both cases, the largest amounts of greenhouse gases were also produced by cows. Milk from pasture farming had an 18% lower carbon footprint (0.99 kg $CO_2 \cdot$kg milk$^{-1}$) than milk from the second type of farming (1.22 kg $CO_2 \cdot$(kg milk$^{-1}$). The difference between the carbon footprints of cheese was 11%, with a footprint of cheese from pasture milk being 15 kg $CO_2 \cdot$(kg$^{-1}$), and of the other cheese 16.9 kg $CO_2 \cdot$(kg$^{-1}$). Considering these data and the fact that the cheese was produced in the same way regardless of the type of milk, it was found that the method of obtaining raw material, in this case the method of breeding dairy cattle, significantly affects the environmental index of the finished product [59].

Liao et al. [60] compared spreads, including 212 plant-based and 40 dairy products, available on markets in 21 European and North American countries. The average carbon footprint of plant products was much lower and amounted to 3.3 kg $CO_2$ per kg of product, while for butter this indicator fluctuated around 12.1 kg $CO_2 \cdot$(kg$^{-1}$), although its sizes were different depending on the country of origin of the product. The highest emissions associated with butter production were recorded in Portugal, Spain, and Greece, while the lowest were in Denmark, Sweden, and Finland. It was also observed that some animal products with significantly reduced fat content were comparable to high-fat plant products. The results were determined by the recipe composition, especially the type of plant fat used, and geographical factors, which largely influenced the environmental footprint determined for the tested animal products.

An important branch of milk processing is the production of dairy food for children and babies. The carbon footprint of producing 1 kg of modified milk was estimated at about 4 kg $CO_2$. In 2012, total sales of these products in six countries (Australia, China, Philippines, India, South Korea, and Malaysia) amounted to 720,450 tonnes. The production of this quantity of food for children corresponds to the emission of landfilling 1 million tonnes of waste, consumption of 323 million gallons of gasoline, or burning 1.4 million tonnes of coal. According to data collected by WHO at the beginning of 2019, it was estimated that the carbon footprint of the products in question increased and oscillated between 11 and 14 kg $CO_2 \cdot$(kg$^{-1}$). The diet of infants and young children is a controversial topic in many communities. Both breastfeeding and the use of preparations replacing breast milk have their supporters and opponents, but from an ecological point of view feeding babies with breast milk has a much smaller impact on the environment. Breastfeeding for half a year allows the reduction of the consumption of industrial preparations for children by 21 kg and reduces greenhouse gas emissions by more than 200 kg $CO_2$ [61].

The carbon footprint (CF) for pork production was determined using life cycle analysis (LCA), which included elements such as breeding pigs, slaughter, retail, and consumption of fresh meat. Pig farming has proved to be the most emissive stage, as it is time and energy consuming (it lasts the longest of all stages of meat production), requires a large amount of feed, and significant amounts of manure and methane are produced. The carbon footprint of pig farming was 4.383 kg $CO_2 \cdot$(kg meat$^{-1}$), which represents over 90% of total emissions. In addition, 95.4% of the emitted nitrogen compounds and 98.4% of sulfur compounds originate from this stage. On this basis, it was found that in order to

reduce the degrading impact of pig production on the environment, one should focus on improving animal husbandry systems [62].

Due to the negative impact of meat production on the environment, Spain has been considering ways to reduce this occurrence. One of the methods proposed was to impose taxes on certain articles of animal origin, including meat, fish, and eggs. Of the products studied, lamb (22.96 kg $CO_2 \cdot (kg^{-1})$) and beef (18.21 kg $CO_2 \cdot (kg^{-1})$) had the largest carbon footprint. Indicators of other products, i.e., turkey, pork, chicken, eggs, and fish were several times smaller and amounted to 5.56, 4.97, 4.02, 3.03, and 2.83 kg $CO_2 \cdot (kg^{-1})$, respectively. With the help of specialized programs and data on consumption and the carbon footprint of the products studied, potential changes in the total value of the carbon footprint were analyzed if different taxes were imposed on selected items. Unexpectedly, the biggest reduction in the total carbon footprint resulted from the imposition of a tax on fish, which was directly related to the amount of average consumption of these products by consumers. The results showed that taxation of the most burdensome products is not always a good solution. It was found that imposing taxes on foodstuffs can significantly contribute to slowing down climate change, but their amount and type of food that will be affected by them depends mainly on the preferences of consumers of a country or region [63].

As mentioned earlier, livestock production, such as beef, consumes enormous amounts of resources such as water and land, which for decades have been harvested by cutting down forests to increase the acreage of agricultural land. Forests play a very important role in the circulation of carbon dioxide in the environment, as they use huge amounts of this gas in the process of photosynthesis, where $CO_2$ is converted into the oxygen necessary for life. In recent years, along with the spread of the principles of sustainable development, the emphasis has been on protecting the Amazonian forests, known as the "green lungs of the planet", and these practices are being curtailed. This phenomenon was particularly pronounced in Brazil, where deforestation has been reduced by more than 70% in recent years. Such activities have an impact on slowing down climate change and improving the quality of the environment [64,65].

The carbon footprint is usually calculated per unit of the product; however, from a nutritional point of view, it can also be calculated per unit of a specific nutrient. In 2012, studies were carried out on the carbon footprint of a protein from various sources. The obtained values ranged from about 4 kg $CO_2 \cdot (kg\ protein^{-1})$ contained in plant meat substitutes, legumes, mussels, and herring to over 600 kg $CO_2 \cdot (kg\ protein^{-1})$ derived from mountain ruminant meat. Large discrepancies have been observed for different categories of animal products, which are associated with the diversity of production systems—each farming method has a different impact on the environment. It was also mentioned that a protein with a higher nutritional value is even 150 times less burdensome for the environment than a protein with a lower value [15].

Research was conducted to compare the carbon footprints of Spanish oranges exported to the European market in the 2012–2013 season. The subject of the experiment were oranges from 21 organic orchards and 21 traditional orchards, including their cultivation, harvesting, preparation for sale, export, and disposal of packaging waste. It turned out that the total carbon footprint of organic oranges is 18% lower than that of traditionally grown oranges. The difference was mainly due to the use of other cultivation systems. Emissions from organic orchards were 52.5% lower, which was associated with the use of only natural fertilizer and a significant reduction in the use of pesticides, because the other stages of the orange's life cycle, such as packaging, transport, and distribution, are the same regardless of the origin of the fruit. In both cases, it was observed that the carbon footprint decreased with the increase in orange harvest per hectare, although in organic farming this trend was more pronounced [66].

Due to the large share of mango production in the Brazilian agri-food sector, an attempt was made to estimate the average annual carbon footprint during three stages: the period of plant growth from planting to the beginning of fruiting (about 5 years), proper production (6–30 years), and fruit packaging. The average carbon footprint was 0.13 kg $CO_2 \cdot (kg\ fruit^{-1})$, although the values of this

indicator ranged from 0.06 to 0.18 kg kg $CO_2 \cdot (kg^{-1})$. Most greenhouse gases (73%) were generated at the stage of tree cultivation and proper production, due to the use of fertilizers and high energy consumption of plant irrigation systems, followed by packaging, which accounted for 23% of total emissions. The possibilities of reducing the carbon footprint of fruits were also examined, among which the most effective proved to be the complete replacement of traditional sources of electricity with renewable ones, such as windmills or solar panels, which could result in a reduction of emissivity by up to 36% [67].

To extend the shelf life of fruits, various chemical and thermal fixation methods are used. Chinese scientists compared the environmental impact of two plum drying methods often used in those areas: infrared-assisted drying with superheated steam and solar assisted by air heating. In both cases, plums were previously washed and osmotically dehydrated, and packed in polyethylene bags after proper drying and delivered to stores. The factors differentiating both plum production processes are the choice of drying method, sugar and energy consumption, and the distance from the courts to the factory and then to the places of distribution. The data used in the study pertained to the 2015–2017 period. The carbon footprint of steamed plums was 5.50 kg $CO_2 \cdot (kg$ dried fruit$^{-1})$, while that of the traditional drying was 5.96 kg $CO_2 \cdot (kg$ dried fruit$^{-1})$. The obtained results were influenced by a greater demand for energy and water, necessary for solar drying plums, and several times greater distance that the raw material had to cover from the orchard to the dryer and from the dryer to the recipient. In both cases, the best solution to reduce the negative impact on climate change would be to replace energy, mainly from coal and oil, with wind energy [68].

The Robusta coffee production process in Thailand was also examined. It was divided into three main stages—the first was the cultivation of plants until harvest, when the green coffee bean was obtained, the second involved the processing from accepting the raw material to the plant to getting roasted grain, while the third stage was the grinding of roasted coffee, followed by the final product. The estimated carbon footprint was about 0.4 kg $CO_2 \cdot (kg$ raw beans$^{-1})$, 0.55 kg $CO_2 \cdot (kg$ roasted coffee$^{-1})$, and 0.56 kg $CO_2 \cdot (kg$ ground coffee$^{-1})$. The cultivation stage (71%) has the largest share in the greenhouse gas emission in the process of obtaining ground coffee, which is mainly due to the need to use various types of fertilizers. Roasting and milling stages are much less burdensome for the environment and are responsible for 27% and 2% of total emissions respectively, which are related to the energy expenditure and water consumption necessary to obtain the products of each of these stages [69].

Coffee is a drink eagerly consumed by people around the world. It can be prepared in many ways and each of them has a different impact on global warming. The carbon footprint of 50 mL of black coffee, including its cultivation, processing, transport (approx. 720 km), preparation, and utilization of waste, is at the level of 11.4 kg $CO_2 \cdot (50$ mL$^{-1})$ for a drink prepared in a French press brewer, of 19.7 kg $CO_2 \cdot (50$ mL$^{-1})$ for coffee brewed at home with heated water on a gas stove, and of up to 35.6 kg $CO_2 \cdot (50$ mL$^{-1})$ for coffee from a capsule espresso machine, where 46% of emissions come from the cultivation of beans and 36% during packaging [70].

The most popular hot drink in the world is tea. The carbon footprint of five tea varieties was assessed, ranging from the field to the distribution of finished products and the emissions from the process of preparing and consuming tea according to traditional proportions (2 g of tea was brewed in 250 mL of boiling water). The carbon footprint of the studied teas from cradle to supermarket gate was 19.2 kg $CO_2 \cdot (kg$ green tea Wuyangchunyu$^{-1})$, 19.9 kg $CO_2 \cdot (kg$ green tea Longjing$^{-1})$, 11.9 kg $CO_2 \cdot (kg$ black tea Wuyangkungfu$^{-1})$, 6.6 kg $CO_2 \cdot (kg$ oolong tea Jinkengoolong-1), and 4.5 kg $CO_2 \cdot (kg$ export teabag green tea$^{-1})$. The emission of $CO_2$ equivalent per cup of tea from cradle to grave was much higher and amounted to 59.4, 63.5, 47.4, 36.9, and 34.5 g $CO_2 \cdot (250$ mL$^{-1})$, respectively, while the CF of the brewing and consumption process itself was equal to 25.7 g $CO_2 \cdot (250$ mL$^{-1})$ for all variants and this accounted for 40–70% of the total emissions [71]. Other published papers proved that average carbon footprint is approximately 32 kg $CO_2 \cdot (kg$ dried tea$^{-1})$, but it can balance from 6 to even 200 g $CO_2 \cdot (cup$ of tea$^{-1})$. Typically, a much higher CF characterizes more teabags (64 g $CO_2 \cdot (cup$ of

tea$^{-1}$)) than loose teas (about 20 g $CO_2$·(cup of tea$^{-1}$)). The main factors influencing these values are methods of cultivating, processing, shipping, packaging, brewing, and discarding [72].

Other studies compared the environmental impact of different coffee growing methods in Vietnam. The first and the most popular method in that region is conventional cultivation, ensuring high yields and high profits, which uses only chemical fertilizers and pesticides. This method is effective, but gives a product with a high content of chemical residues and low quality. The second method was defined as conventional, in which synthetic fertilizers and plant protection products were replaced by natural ones. The last method was organic farming, safe for the environment and providing products of high quality and nutritional value. The data collection period was 30 years and began at the time of planting, including cultivating coffee beans, harvesting them, drying them, removing the skin, and packing in bags. As expected, the conventional method had the most negative impact on the environment, and organic farming the least. The carbon footprints were adequately: 0.920–0.949, 0.721–0.736, and 0.640–0.647 kg $CO_2$·(kg dried coffee beans$^{-1}$). The use and type of fertilizers and plant protection products used have been identified as the main cause of these differences [73].

Pishgar-Komeleh et al. [74] determined the average carbon footprint of tomato cultivation in Iran at 0.26 kg $CO_2$·(kg$^{-1}$). The results were comparable with other studies, and the differences were due to the distance over which the vegetables had to be transported from the plantation to the place of distribution. The farm had abandoned traditional irrigation and used a modern electric droplet system. Due to the large demand for water for tomatoes, the energy consumption of this system had the largest share in total greenhouse gas emissions. In order to optimize the indicators, models were used, according to which the use of modern irrigation systems, as well as the abandonment of artificial fertilizers and plant protection products in favor of natural, could reduce the carbon footprint by up to 43% with a slight increase in energy consumption. The environmental footprint of cherry tomato production in Tunisia, including energy consumption, fertilization, pesticides, transport, greenhouse maintenance, and waste management, was 0.954 kg $CO_2$·(kg$^{-1}$), of which about 80% was due to the use of large amounts of energy [75].

In 2015, in the United States of America, the environmental footprint of tomato preserves (concentrate and diced tomatoes) was estimated at 0.827 and 0.157 kg $CO_2$ per kg product. The differences were due to the use in production processes, unit operations (thermal processes (evaporation) are more energy consuming than mechanical processes (dicing)), and raw material efficiency (6 kg of tomatoes was needed to produce 1 kg of concentrate, but only 1.3 kg of tomatoes to produce 1 kg of diced preserve). In 2005 greenhouse gases emissions were equal to 0.945 and 0.213 kg $CO_2$·(kg$^{-1}$), so these values decreased by 12% and 26% for last 10 years [76].

It turns out that not only the production, but also the method of managing food waste is of key importance in the amount of greenhouse gases emitted in the product's life. The Walmart chain of stores in Mexico has conducted comparative studies on the carbon footprint of various food waste disposal methods. The total amount of waste emitted by the market chain in 2017 was over 377 million tons, of which food was the most difficult material of which to dispose. For research purposes, food waste, i.e., spoiled products, with an exceeded shelf life and determined by employees as unfit for consumption, was divided into six categories: meat products, bread, fruit and vegetables, dairy products, non-meat products, and mixed food. The basic method of managing the abovementioned waste was landfilling and composting, which underwent less than 5% of the total amount of waste, mainly fruit and vegetables. Alternative methods that fit into the zero waste policy included improving food management, returning products with a close expiry date to food banks, and biogas production (about 56% of waste). It has been estimated that alternative waste management has reduced greenhouse gas emissions by approximately 135,301 Mt $CO_2$, which corresponds to annual emissions from the use of 28,484 passenger cars [77].

One of the key residues from rice cultivation is large amounts of straw, which is usually burned in fields (greenhouse gas emissions emissions about 4 kg $CO_2$·(t dry straw$^{-1}$). Scientists from India have developed a method of producing ethanol from materials such as rice and wheat straw, cotton and

soybean stalks, sugar cane pomace, and corn. This method was promising enough that it can be used on an industrial scale. It is based on the fractionation of cellulose into fractions of different chain lengths, subjecting them to enzymatic hydrolysis to obtain glucose, which can be used as a nutrient for yeast. The carbon footprint of ethanol obtained by this method was equal to 2.82 kg $CO_2$ L alcohol$^{-1}$. About 86% of emissions are caused by high energy consumption, which in India comes mainly from fossil sources. The effect of using rice straw as a material for ethanol production, combined with the change of the energy source necessary to carry out the process to a more ecological one, may be the reduction of total greenhouse gas emissions by about 0.394 kg $CO_2$ per L of ethanol [78].

## 3. Conclusions

The concept of sustainable development is increasingly important and is successively introduced into the political and economic activities of countries around the world. It is an incentive not to take away from people the goods necessary to meet every day needs today and in the future.

The agri-food sector is the branch of the economy that has the greatest impact on the environment and is associated with a high demand for energy, water, land, and chemicals supporting agricultural production and packaging. Actions are carried out to thoroughly understand the causes and reduce this occurrence as much as possible. To this end, tools such as life cycle assessment of products and services are used to assess environmental indicators (footprints). The most commonly used include carbon, water, and energy footprints that determine greenhouse gas emissions and the inputs necessary to produce a product. When calculating foodstuffs, all production stages are taken into account, starting with the cultivation of plants or animals through harvesting, processing, distribution, and ending with waste treatment. The value of these indicators is influenced by many factors related to the specificity of the production process, location, environmental conditions, and many others, which is why there may be very large differences between the indicators of the same products.

The results of the research carried out so far clearly show that the cultivation of plants is much less harmful to the environment compared to the production of meat and other animal products, which is associated with the complexity of the entire production process. The carbon footprint of meat is from several to several dozen times greater than that of most fruit and vegetables. From this perspective, it seems beneficial to reduce the consumption of meat and animal products. One of the goals of sustainable development is to reduce the use of the Earth's natural resources, so it should be considered whether the amount of livestock production should be reduced and replaced by plant cultivation.

Investigating the impact of the agri-food industry on the environmental footprints, especially the carbon footprint, is becoming more common. This is very important, because the analysis of techniques used to manufacture products allows the identification of the weakest links in the production chain and gives them the opportunity to improve. Therefore, it is necessary to continue research in this area and extend it to the largest possible number of products.

**Author Contributions:** The following statements should be used Conceptualization, M.K., A.C., A.L. and M.J.; methodology, M.K., M.J. formal analysis, M.K., A.C., A.L. and M.J.; investigation, M.K., A.C., M.J.; resources, M.K., M.J.; data curation, M.K., M.J.; writing—original draft preparation, M.K., M.J.; writing—review and editing, M.K., M.J.; visualization, M.K., M.J.; supervision, A.C., A.L. and M.J.; project administration, M.J.; funding acquisition, A.C., A.L. and M.J.; All authors have read and agreed to the published version of the manuscript.

**Funding:** This work was founded by the National Center for Research and Development, as part of the III BIOSTRATEG—BIOSTRATEG3/343817/17/NCBR/2018.

**Acknowledgments:** This work was founded by the National Center for Research and Development, as part of the III BIOSTRATEG. "The development of an innovative carbon footprint calculation method for the basic basket of food products" task in the project "Development of healthy food production technologies taking into consideration nutritious food waste management and carbon footprint calculation methodology" BIOSTRATEG3/343817/17/NCBR/2018 was also co–financed by a statutory activity subsidy from the Polish Ministry of Sciences and Higher Education for the Faculty of Food Sciences of Warsaw University of Life Sciences.

**Conflicts of Interest:** The authors declare that they have no known competing financial interests or personal relationships that could have appeared to influence the work reported in this paper.

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
