# Peer review of "Sustainable Development in the Agri-Food Sector in Terms of the Carbon Footprint: A Review"

_sustainability, doi:10.3390/su12166463_

Round 1

Reviewer 1 Report

This review article “sustainable development in the agri-food sector in terms of the carbon footprint” discusses mostly carbon footprints of food production along with water, nitrogen and energy footprints. It’s a well written review but there are issues that need to be addressed.

  1. In Line 156: While discussing water footprints, authors mentioned “water not physically present in the product, but used to make it”. I strongly suggest that authors discuss the example of meat production here. They should write about the gallons of water required to produce per kg of beef, pork etc.
  2. In the section “energy footprints” authors should discuss about the topic of “food miles” and its impact on sustainability.
  3. I’m quite disappointed by not finding any charts or tables in this article. Authors should include at least one chart that depicts carbon footprints based on food category.
  4. Also, it is well known that meat is the number 1 contributor of carbon emission in food category. There was not enough “criticism” of this fact in this article. Also the amount of deforestation has been happening in Amazon for “beef” has been a hot topic worldwide. Is not it related to carbon footprints and sustainability?
  5. Authors discussed about carbon footprints from coffee production but what about tea? Tea is the most popular drink in populous Asian countries. Also I would encourage authors to include carbon footprints of carbonated beverages-a western world favorite. There are reports that revealed that a 330-mL can of cola has a carbon footprint of ~150-175 g. All can be clubbed together as “beverages”.

Author Response

We would like to hanks for reviewers’ valuable comments and recommendation. Suggestions were analysed, all of them were taken into account in the manuscript and marked in red (Reviewer 1) and blue (Reviewer 2) colour. In addition we provide here answers to specific issues.  We have modified the text in accordance with comments and added new references.

A detailed description of the changes introduced in the pdf attachment.

Yours sincerely,

Monika Janowicz

Reviewer 2 Report

Comments and Suggestions for Authors

This paper tried to examine the topics of The concept of sustainable developmentregarding in the agri-food sector and global economy. It must have taken much effort to conduct the research but its contribution is not clear to reader. In specific, it should be improved in the following ways.

1. comments

(1) Contribution
This research is based on environmental footprints of products and services are calculated using the LCA (Life Cycle Assessment) method. That is, please discuss more about the sustainable development and environmental footprints, especially the carbon footprint in the agri-food sector, as well as contribution of this research and explain how the research findings can be used to give values to different area in sustainable development in the agri-food sector.

(2) Methods
Please include more information about why the determined by many factors associated with their production regarding carbon footprint of food products. In this currents analyses, it is still lack of solid explanations in GHG emission reduction of the use of artificial fertilizers and plant protection products.

(3) Theoretical and managerial implications
Please extend the discussions of agri-food sectors’ believed that sustainability practicesin this paper and link them to the contributions. In this paper, the discussions and implications are linked to the empirical findings. Instead of linking to the empirical results, it is recommended to emphasize the sustainability practices in the cities what the previous studies could not identify but this paper did.

(4) Please provide information about 'the structure of the remainder of the paper' at the end of introduction part.

Author Response

(The authors gave the same response as above.)

Round 2

Reviewer 2 Report

All my concerns have been adequately answered;

Best wishes with your future research!